# Alkyl nitrates in the boreal forest: Formation via the NO₃, OH and O₃ induced oxidation of BVOCs and ambient lifetimes

Jonathan Liebmann[1], Nicolas Sobanski[1], Jan Schuladen[1], Einar Karu[1], Heidi Hellén[2], Hannele Hakola[2], Qiaozhi Zha[3], Mikael Ehn[3], Matthieu Riva[3], Liine Heikkinen[3], Jonathan Williams[1], Horst Fischer[1], Jos Lelieveld[1] and John N. Crowley[1]

[1]Division of Atmospheric Chemistry, Max Planck Institut für Chemie, 55128, Mainz, Germany
[2]Finnish Meteorological Institute, 00560, Helsinki, Finland
[3]Institute for Atmospheric and Earth System Research / Physics, University of Helsinki, 00014, Helsinki, Finland
*Correspondence to:* John Crowley (john.crowley@mpic.de)

**Abstract.** The formation of alkyl nitrates in various oxidation processes taking place throughout the diel cycle can represent an important sink of reactive nitrogen and mechanism for chain-termination in atmospheric photo-oxidation cycles. The low volatility alkyl nitrates formed from biogenic volatile organic compounds (BVOCs), especially terpenoids, enhance rates of production and growth of secondary organic aerosol. Measurements of the NO₃-reactivity and the mixing ratio of total alkyl nitrates (ΣANs) in the Finnish boreal forest enabled assessment of the relative importance of NO₃-, O₃- and OH-initiated formation of alkyl-nitrates from BVOCs in this environment. The high reactivity of the forest air towards NO₃ resulted in reactions of the nitrate radical with terpenes contributing substantially to formation of ANs not only during the night but also during daytime. Overall, night-time reactions of NO₃ accounted for 49% of the local production rate of ANs, with contributions of 21%, 18% and 12% for NO₃, OH and O₃, during the day. The lifetimes of the gas-phase ANs formed in this environment were of the order of 2 hours due to efficient uptake to aerosol (and dry-deposition), resulting in the transfer of reactive nitrogen from anthropogenic sources to the forest ecosystem.

## 1 Introduction

Alkyl nitrates (ANs, $R\text{-}CH_2ONO_2$) are formed in chain terminating reactions that limit photochemical cycling of organic and $HO_x$ radicals and represent an important sink for atmospheric nitrogen (Liu et al., 2012; Lee et al., 2016; Huang et al., 2019). During daytime, ANs are formed in a minor branch of the reaction of NO with organic peroxy radicals ($RO_2$) (Lightfoot et al., 1992), which are formed mainly via the OH-initiated oxidation of volatile organic compounds (VOCs):

| | | |
|---|---|---|
| $OH + RH \ (+ O_2)$ | $\rightarrow RO_2$ | (R1) |
| $RO_2 + NO + M$ | $\rightarrow RONO_2 + M$ | (R2a) |
| $RO_2 + NO$ | $\rightarrow RO + NO_2$ | (R2b) |

The yield of ANs in reactions R1-R2 varies with the structure of the R-substituent, temperature and pressure and for small alkyl groups (e.g. R = H, $CH_3$, $C_2H_5$) is generally less than a few percent at 1 bar and 298 K but may increase to values close to 20% for $RO_2$ formed from the OH-initiation of biogenic VOCs (Perring et al., 2013; Lee et al., 2014b; IUPAC, 2019).

The reaction with $O_3$ represents an additional sink for biogenic VOCs (BVOCs) (Peräkylä et al., 2014; Yan et al., 2016) which, in the presence of NO, can also lead to the formation of alkyl nitrates. The reaction proceeds by initial addition of $O_3$ to the C=C bond of e.g. a terpene to form a primary ozonide (POZ, R3) which rapidly decompose (R4) via Criegee intermediates to form OH and (in the presence of $O_2$) $RO_2$. The latter react via R1 and R2 to form alkyl nitrates.

| | | |
|---|---|---|
| $O_3 + R=R$ | $\rightarrow POZ$ | (R3) |
| $POZ \ (+O_2)$ | $\rightarrow OH + RO_2$ | (R4a) |
| $POZ$ | $\rightarrow$ other products | (R4b) |

At night-time, OH radical concentrations are very low and the $NO_3$ radical, formed by the oxidation of $NO_2$ by ozone (Reaction R5), is the main initiator of oxidation of several classes of VOCs with Reaction (R6a) the dominant pathway for alkyl nitrate formation (Rosen et al., 2004; Crowley et al., 2010; Rollins et al., 2013; Sobanski et al., 2017).

| | | |
|---|---|---|
| $NO_2 + O_3$ | $\rightarrow NO_3 + O_2$ | (R5) |
| $NO_3 + VOC \ (O_2, NO, RO_2, HO_2)$ | $\rightarrow AN$ | (R6a) |
| $NO_3 + VOC$ | $\rightarrow$ other products | (R6b) |

Reaction 6a is a composite process involving initial formation of a nitroalkyl radical (via electrophilic addition of $NO_3$ to a C=C double bond) followed by the formation of a nitrooxyperoxy radical (via further addition of $O_2$) which can react with NO, $NO_3$, $HO_2$ or $RO_2$ to form substituted alkyl-nitrates (IUPAC, 2019).

The branching ratio to AN formation via $NO_3$ oxidation is generally much larger than that for organic peroxy radicals reacting with NO and for biogenic VOCs (BVOCs) can approach 80% (Ng et al., 2017; IUPAC, 2019). The formation of ANs via the degradation of saturated and unsaturated VOCs initiated by $NO_3$, OH and $O_3$ are summarized in Fig. 1.

During daytime, the reactions of $NO_3$ with VOCs are often reduced in importance by the rapid photolysis of $NO_3$ and by its reaction with NO (R7-R9)(Wayne et al., 1991).

| NO$_3$ + NO | $\rightarrow$ 2 NO$_2$ | (R7) |
| NO$_3$ + $h\nu$ | $\rightarrow$ NO$_2$ + O | (R8) |
| NO$_3$ + $h\nu$ | $\rightarrow$ NO + O$_2$ | (R9) |

The fraction (*f*) of NO$_3$ radicals that undergo reaction with VOCs can be calculated according to Eq. 1

$$f = \frac{k_{OTG}^{NO_3}}{k_{OTG}^{NO_3} + J_{NO_3} + k_7[NO]} \tag{1}$$

where $k_{OTG}^{NO_3}$ is the first-order loss-frequency for NO$_3$ reaction with organic trace gases (NO$_3$-reactivity), $J_{NO_3}$ is the photolysis rate constant and $[NO]k_7$ the NO concentration multiplied by the rate constant for Reaction (R7). Recent measurements of NO$_3$-reactivity in forested regions (Ayres et al., 2015; Liebmann et al., 2018a; Liebmann et al., 2018b) suggest that, even during the day, a significant fraction of the nitrate radicals generated in Reaction (R5) can react with BVOCs rather than undergoing photolysis or reaction with NO to re-form NO$_x$ (NO$_x$ = NO + NO$_2$). During a 2016 field intensive in the boreal forest (IBAIRN, Influence of Biosphere-Atmosphere Interactions on the Reactive Nitrogen budget), we showed that, on average more than 20% of NO$_3$ radicals formed during the day were lost due to reaction with BVOCs (Liebmann et al., 2018a). In this work, we examine the contribution of the NO$_3$–initiated oxidation of VOCs to the formation of ANs both during the day- and night-time during IBAIRN and compare this to AN formation initiated by reactions of OH and O$_3$. Using calculations of the overall production rate of ANs and measurements of the summed mixing ratio of ANs (ΣANs), we derive a lifetime for ANs in this environment.

## 2 Measurements

The measurements were made during the IBAIRN field intensive in September 2016 at the "Station for Measuring Forest Ecosystem-Atmosphere Relations II" (SMEAR II) in Hyytiälä (61°51´N, 24°17´E) in southern Finland. A detailed description of the measurement site can be found elsewhere (Rinne et al., 2005; Lappalainen et al., 2009; Aaltonen et al., 2011). Briefly, the measurement site is located in the boreal forest with mostly biogenic influences. Anthropogenic emissions from two larger cities (Tampere and Jyväskylä) and a local sawmill occasionally impacted the site, the former resulting in an increase in NO$_X$ levels, the latter in BVOCs.

The IBAIRN campaign took place in the transition between summer and autumn with the length of day shortening from 14.0 h at the beginning of the campaign to 11.5 h at the end. Campaign temperatures, relative humidity, wind-speed and direction can be found in Fig. S1 of the supplementary information. During IBAIRN, the diel temperature varied from a night-time minimum of 2°C to a daytime maximum of 20°C, with the nights often characterized by a strong temperature inversion and a very shallow boundary layer, which resulted in higher monoterpene mixing ratios than during daytime. The relative humidity reached 100% during many nights with ground-level fog formation. There was no rainfall during the campaign and the wind was predominantly from north-western directions with wind-speeds never exceeding 5 m s$^{-1}$. The NO$_x$ levels during the entire campaign were low (mean value of 320 pptv) with occasional increases (up to 1.4 ppbv) when the site experienced air

masses with trajectories that passed over urban centres. Daily $O_3$ maxima were 30-35 ppbv, with much lower values (5-10 ppbv) on those nights with strong temperature inversions (Liebmann et al., 2018a).

The mixing ratios of OH, NO, $NO_2$, $O_3$, $\Sigma$ANs, VOCs, the $NO_3$ photolysis frequency, and the $NO_3$-reactivity are required to examine the formation and loss of ANs during IBAIRN. Most instruments used for the data analysis sampled from a common inlet (at 8 m height) on a clearing within the boreal forest. Exceptions were measurement of some organic trace gases and the measurements of actinic flux (see below). As the instruments have been described previously (Liebmann et al., 2018a), we list only the limits of detection (LOD) and $2\sigma$ total uncertainty here.

NO was measured using a modified commercial chemi-luminescence detector (LOD = 5 pptv in 60 s, $2\sigma$ = 20%). $O_3$ was measured by optical absorption (LOD = 1 ppbv in 10 s, $2\sigma$ = 5%). $NO_2$ (LOD = 60 pptv in 6 s, $2\sigma$ = 6%) and $\Sigma$ANs (LOD = 40 pptv in 10 min, $2\sigma$ = 20%) were measured using the 5-channel, thermal dissociation-cavity ring down spectrometer (TD-CRDS) described in detail by Sobanski et al. (2016). The TD-CRDS instrument also indicated that $NO_3$ levels were < 1 pptv throughout the campaign.

$J_{NO3}$ was calculated from actinic flux measured at 35 m height using a spectral radiometer (Metcon GmbH) and evaluated $NO_3$ cross sections / quantum yields (Burkholder et al., 2015). Ultraviolet-B radiation (280-320 nm) was sampled at 18 m height (Solar Light SL501A radiometer). $NO_3$-reactivity was measured with a recently developed instrument coupling a flow-tube reactor with CRDS (Liebmann et al., 2017; Liebmann et al., 2018a; Liebmann et al., 2018b). Isoprene and monoterpenes were measured from a common inlet using a Gas-Chromatograph/Atomic emission detector (GC-AED, LOD 1 pptv in 20 min, $2\sigma$ 14%), as described in (Liebmann et al., 2018a). Organic acids, alcohols, aldehydes as well as several alkanes were measured on the same clearing but at 1.5 m height, roughly 30 m away from the common inlet using a Gas-Chromatograph / Mass Spectrometer set up (GC-MS). Details are found in Hellén et al. (2018). OH radical concentrations were not directly measured during IBAIRN but obtained from a correlation of ground-level OH-measurements with ultraviolet B radiation intensity ([UVB], in units of W m$^{-2}$) at this location with $[OH] = 5.62 \times 10^5$ $[UVB]^{0.62}$ (Rohrer and Berresheim, 2006; Petäjä et al., 2009; Hellén et al., 2018). The calculated, ground level OH concentrations were multiplied by a factor 2 to take gradients in OH between ground-level and at canopy height into account (Hens et al., 2014). The OH concentrations have an associated uncertainty of ~50%.

Real-time measurements of gas- and particle phase oxidation products formed in the boreal forest were conducted using an Aerodyne high-resolution, long time-of-flight chemical ionization mass spectrometer (HR-L-ToF-CIMS), equipped with iodide ($I^-$) reagent ion chemistry (Lopez-Hilfiker et al., 2014; Lopez-Hilfiker et al., 2015; Lee et al., 2016; Riva et al., 2019). The instrument was coupled to a Filter Inlet for Gases and AEROsols (FIGAERO). Analyses were restricted to ions containing an iodide adduct, which guarantees detection of the parent organic compounds without substantial fragmentation. Iodide-CIMS has been described previously and demonstrated high sensitivity towards oxygenated organic compounds including alkyl nitrates both in the gas and particle phases (Lee et al., 2016). Aerosol surface areas were calculated using data from a differential mobility particle sizer permanently installed at the SMEAR station.

## 3 Results and discussion

### 3.1 ANs production from NO₃ reactions with VOCs

In Fig. 2 we display a time series of the $NO_3$ precursors ($NO_2$ and $O_3$) and the $NO_3$-reactivity ($k_{OTG}^{NO_3}$) together with the $NO_3$ loss rate constant resulting from its reaction with NO and photolysis. The latter also serves to delineate day ($J_{NO3} \geq 5 \times 10^{-4}$ s$^{-1}$) and night.

The instantaneous production rate of ANs from the reaction of $NO_3$ with VOCs ($\sum P_{ANs}^{NO_3}$) is given by:

$$\sum P_{ANs}^{NO_3} = \overline{\alpha}^{NO_3} f [O_3][NO_2] k_5 \tag{2}$$

where $\overline{\alpha}^{NO_3}$ is an average AN-yield. Assuming that all the VOCs responsible for loss of $NO_3$ were identified and quantified the average yield can be derived (Eq. 3) from VOC- specific values of $\alpha_i^{NO_3}$ weighted by their relative contribution to $k_{OTG}^{NO_3}$.

$$\overline{\alpha}^{NO_3} = \frac{\sum \alpha_i^{NO_3} k_i^{NO_3}[C_i]}{k_{OTG}^{NO_3}} \tag{3}$$

where $[C_i]$ is the concentration of the specific VOC. The rate constants and branching ratios used to calculate $\sum P_{ANs}^{NO_3}$ for individual BVOCs can be found in Table S1 of the supplementary information. A large selection of VOCs was measured (a listing is given in the caption to Figure S2 of the supplementary information) but the 5 biogenic VOCs listed (α-pinene, β-pinene, carene, limonene, isoprene) accounted for > 98 % of the attributed $NO_3$ reactivity.

Combining expressions (1-3), we derive:

$$\sum P_{ANs}^{NO_3} = \frac{\sum \alpha_i^{NO_3} k_i^{NO_3}[C_i]}{k_{OTG}^{NO_3} + J_{NO_3} + k_7[NO]} [O_3][NO_2] k_5 \tag{4}$$

Recognizing that $[O_3][NO_2] k_5$ and the term ($k_{OTG}^{NO_3} + J_{NO_3} + k_7[NO]$) are the total $NO_3$ production rates and loss rates, respectively, and that their ratio is the concentration of $NO_3$ in steady state, $[NO_3]_{ss}$, we can also write:

$$\sum P_{ANs}^{NO_3} = [NO_3]_{ss} \sum \alpha_i^{NO_3} k_i^{NO_3}[C_i] \tag{5}$$

Expression (5) can be used to calculate the production rates of ANs if $NO_3$ measurements are above the detection limit, which despite deployment of sensitive instrumentation, was not the case in IBAIRN or in previous campaigns in the boreal forest (Rinne et al., 2012; Crowley et al., 2018). This highlights the advantage of measuring the overall reactivity of $NO_3$ instead of its concentration in highly reactive environments.

As described by Liebmann et al. (2018a) the VOC measurements did not account for the total measured reactivity. The reactivity that could not be attributed ($k_{unattributed}$, see eq. 6) (on average 30% at night-time and 60% during daytime) was therefore treated as stemming from a VOC with an alkyl nitrate yield of 0.7.

$$k_{unattributed} = k_{OTG}^{NO_3} - \sum k_i^{NO_3}[C_i] \tag{6}$$

A value of 0.7 was chosen as the unattributed reactivity is likely to be due to highly reactive BVOCs (e.g. terpenes that were not measured or sesquiterpenes, see Liebmann et al. (2018a)), which have alkyl nitrate yields between 0.6-0.8 (IUPAC,

2019). The uncertainty related to the calculation of $\sum P_{ANs}^{NO_3}$ is estimated as 65% with contributions of 50% from $(k_{OTG}^{NO_3} + J_{NO3} + k_7[NO])$, 18% from $[O_3][NO_2]k_5$, 30% from $\alpha_i^{NO_3}$, 15% from $k_i^{NO_3}$ and 15% from $[C_i]$.

## 3.2 ANs production from OH reactions with VOCs

The rate of production of ANs from the OH radical initiated oxidation of VOCs ($P_{ANs}^{OH}$) in the presence of NO can be calculated using Eq. 7:

$$\sum P_{ANs}^{OH} = [OH] \sum \alpha_i^{RO_2} k_i^{OH}[C_i] \tag{7}$$

where [OH] is the OH concentration, $\alpha_i^{RO_2}$ the VOC-specific branching ratio for alkyl nitrate formation (via R2a) from the peroxy radical ($RO_2$) formed and $k_{OTG}^{OH}$ is the OH-reactivity derived from the VOC concentrations $[C_i]$ and the corresponding rate constant $k_i^{OH}$ for reaction with OH.

Organic peroxy radicals formed in reaction R1 or R4a do not react solely with NO but can also react with hydroperoxyl radicals (R10) and undergo radical recombination (R11) each of which reduces the production rate of ANs via Reaction (R2a). They can also isomerize (R12) to form a more oxidized form of $RO_2$.

$$RO_2 + HO_2 \quad \rightarrow ROOH + O_2 \tag{R10}$$

$$RO_2 + RO_2 \quad \rightarrow products \tag{R11}$$

$$RO_2 \text{ (isomerisation)} + O_2 \rightarrow R'O_2 \tag{R12}$$

Although the rate constant for some cross- and self-reactions of terpene derived $RO_2$ are large (Berndt et al., 2018), the organic peroxide products of these reactions have only been observed in very low mixing ratios at this site (Yan et al., 2016) and, following Browne et al. (2013), we neglect the impact of Reaction (R11). We also assume that the peroxy radical formed in R12 forms an organic nitrate with the same efficiency as the parent $RO_2$, so that R12 can also be neglected. The $RO_2$ formed in (R1) and (R4a) do not form stable peroxy-nitrates in their reaction with $NO_2$, so this $RO_2$ loss process can be safely neglected.

To assess the influence of reaction R10, we used the noon-time ratio of OH to $HO_2$ radicals derived by Crowley et al. (2018) during a campaign at the same location ($HO_2 \approx 250 \times OH$) to calculate the $HO_2$ concentration during IBAIRN from that of OH (itself calculated from actinic flux, see above). The fraction, $\beta$, of the $RO_2$ radicals that react with NO is then:

$$\beta = \frac{k_{RO_2+NO}[NO]}{k_{RO_2+NO}[NO] + k_{RO_2+HO_2}[HO_2]} \tag{8}$$

$\beta$ was derived by assuming a generic rate coefficient (based on rate coefficients of known $RO_2$ (IUPAC, 2019)) of $k_{RO_2+NO} = 8 \times 10^{-12}$ cm$^3$ molecule$^{-1}$ s$^{-1}$ and $k_{RO_2+HO_2} = 1 \times 10^{-11}$ cm$^3$ molecule$^{-1}$ s$^{-1}$. Campaign average values of $\beta$ varied from $\approx 0.9$ in the early morning (04:00 - 08:30 UTC) during the breakup of the night-time boundary layer, decreasing slightly to $\approx 0.8$ later in the day (09:30-16:30 UTC). In order to account for this competition with $HO_2$ reactions, Equation (7) can be modified to:

$$\sum P_{ANs}^{OH} = [OH]\beta \sum \alpha_i^{RO_2} k_i^{OH}[C_i] \tag{9}$$

The rate constants and branching ratios used for these calculations can be found in Table S1 of the supplementary information. The total OH-reactivity ($k_{OTG}^{OH}$) was calculated using the measured concentrations of monoterpenes and isoprene as other VOCs (e.g. aldehydes, acids, alkanes, alkenes) contributed less than 6% (Fig S2). The calculated OH-reactivity from monoterpenes varied between approximately 0.2 and 2 s$^{-1}$ with a campaign average of $\approx$ 0.3 s$^{-1}$, in accordance with previous measurements (Hellén et al., 2018). The high value of 7 s$^{-1}$ on 09.09 was associated with large BVOC mixing ratios in air masses originating from the local sawmill. The calculated production rate of ANs via Equation (9) assumes that all ambient VOCs that result in AN formation were measured. However, previous summertime comparisons of total OH-reactivity derived from VOC measurements suggests that a significant fraction of VOC reacting with OH were not identified (Nölscher et al., 2012) and that this fraction depended on the degree of heat and drought induced stress, with 58% of the measured OH-reactivity remaining unassigned to individual VOCs during "non-stressed" conditions. As OH-reactivity data is not available during IBAIRN we cannot assess which fraction of OH-reactivity could potentially be missing for the much cooler autumn conditions, but expect this to be less than the 58% reported for the warmer summer months. We show below that even if unattributed OH-reactivity reaches 50%, this would not significantly modify our conclusions. We estimate an uncertainty in the term $\sum P_{ANs}^{OH}$ of $\approx$ 70% with contributions of 50% from [OH], 30% from β, 15% from $k_i^{OH}$, 30% from $\alpha_i^{RO_2}$ and 25% from [C$_i$].

### 3.3 ANs production from O$_3$ reactions with VOCs

The production rate of alkyl nitrates generated via the reaction of O$_3$ with VOCs ($\sum P_{ANs}^{NO_3}$) in the presence of NO is described by Equation 10.

$$\sum P_{ANs}^{O_3} = [O_3]\beta \sum \alpha_i^{O_3} k_i^{O_3}[C_i] \alpha_i^{RO_2} \tag{10}$$

where [O$_3$] is the O$_3$ concentration, $\alpha_i^{O_3}$ is the VOC specific yield of the RO$_2$ radical formed in R4, $\alpha_i^{RO_2}$ the VOC specific yield of ANs from RO$_2$ + NO and is assumed to be the same as for OH-initiated degradation of the same BVOCs, $k_{OTG}^{O_3}$ is the O$_3$-reactivity derived from the VOC concentrations [C$_i$] along with the corresponding rate constant $k_i^{O_3}$ for reaction with O$_3$. Similar to OH-reactions, β is the fraction of RO$_2$ radicals that react with NO rather than with HO$_2$ or with themselves (see above).

The OH radicals that are formed in reaction 4a will react according to reactions 1-2 and would also increase the alkyl nitrate yield. However, the calculated OH concentration, based on an empirical correlation of observed [OH] with the ultraviolet B radiation contains all OH sources, including reaction R4a. In order to avoid double-counting this source of OH, the RO$_2$ formed in the sequence R4, R1, R2 are not taken into account in Equation (10). The rate constants and branching ratios used for these calculations can be found in Table S1 of the supplementary information.

We cannot assess which fraction of $O_3$-reactivity could potentially be missing, hence the obtained production rates have to be considered a lower limit. The calculated $O_3$-reactivity is depicted in Fig. S2 of the supporting information. While OH molecules are only abundant in higher concentrations during day, $O_3$ is an important oxidizing agent present during day and night. At night, away from direct sources, the NO mixing ratio is reduced to very low values within a few minutes after sunset due to reaction with $O_3$ to form $NO_2$ and therefore the alkyl nitrate production via this channel becomes insignificant. We used the $NO_3$ photolysis rate to delineate day and night and assumed that in the absence of light ($J_{NO3} < 5 \times 10^{-4}$ s$^{-1}$) the NO mixing ratios are not sufficient to support production of alkyl nitrates. The uncertainty in the term $\sum P_{ANs}^{O_3}$ was estimated to be ~60% with contributions of 5% from [$O_3$], 30% from $\beta$, 30% from $\alpha_i^{O_3}$, 15% from $k_i^{O_3}$, 30% from $\alpha_i^{RO_2}$ and 15% from [$C_i$].

## 3.4 Relative importance of OH, $O_3$ and $NO_3$ initiated VOC oxidation reactions for AN formation

The $NO_3$ production rate, the OH mixing ratios and the calculated alkyl nitrate production rates from $NO_3$, OH and $O_3$ are depicted in Fig. 3. In this low-NOx environment, the $NO_3$ production rate is highly correlated with $NO_2$ mixing ratios, while the OH mixing ratio is directly coupled to solar radiation. The rate of production of ANs from ozonolysis of BVOC has a daytime minimum at noon, with maximum values observed in the late afternoon.

The largest production rates of ANs are observed at night-time via $NO_3$ reactions though $\sum P_{ANs}^{NO_3}$ is highly variable with values between $\approx$ 5 and 65 pptv h$^{-1}$. In contrast, daytime production via OH ($\sum P_{ANs}^{OH}$) is rather reproducible with maximum values of about 21 pptv h$^{-1}$ at local noon. On cloudy days (e.g. 10[th] and 15[th] Sept), the OH production rate is reduced significantly and at the same time the $NO_3$ photolysis rate decreases so that its reactions with VOCs become more important. Both effects combine to enhance the rate of AN generation via $NO_3$ over that of OH. $\sum P_{ANs}^{O_3}$ can reach up to 16 pptv h$^{-1}$ in the late afternoon but is usually between 3-5pptv h$^{-1}$. Figure 3 also plots the time series of $\Sigma$ANs during the campaign. The mixing ratios of $\Sigma$ANs are generally very low (generally < 100 pptv), and are sometimes at (or close to) the instrument's limit of detection.

The campaign-averaged contribution of OH, $O_3$ and $NO_3$ to the production of ANs during IBAIRN is displayed as a diel-profile in Fig. 4a. The night-time generation of ANs formed via $NO_3$ reaction with BVOCs maximises at $\approx$ 20 pptv h$^{-1}$ (at $\approx$ 19:00) which can be compared to the maximum value of just 10 pptv h$^{-1}$ from OH-reactions and 6 pptv h$^{-1}$ from $O_3$-reactions during the day. As almost all $NO_3$ formed in this envoronment will react with a BVOC at nightime, the production rate of ANs is not sensitive to the mixing ratios of BVOCs. The peak in the nightime production rate of ANs at 19:00 coinincides with large $O_3$ and $NO_2$ mixing ratios (Fig 4b), the reduction of both $O_3$ and $NO_2$ between ~ 20:00 and mid-night (UTC) resulting in the decrease in $\sum P_{ANs}^{NO_3}$ during the night, though changes in relative concenrations of the terpenes may also play a role. The daytime generation of ANs from $NO_3$ reactions is significant and at times approaches 50% of the overall rate of AN formation.

The total contributions of OH-, O$_3$- and NO$_3$-induced alkyl-nitrate formation over the whole campaign are summarized in pie-chart form in Fig. 5. In total, 51% of ANs were formed during the day, with 21% initiated by NO$_3$, and 18% initiated by OH chemistry and 12% initiated by O$_3$ chemistry. At night-time the ANs production rate (exclusively via NO$_3$-initiated reactions) contributes 49%.

So far we have assumed that the measured VOCs account for the entire reactivity of the OH radical. If, for example, 50% of the OH-reactivity were not accounted for by the measured VOCs (see section 3.2) the contribution of OH would increase from 18% to 31% with the contributions of NO$_3$ decreasing to 18% (day) and 41% (night). The contribution of O$_3$ would decrease to 10%. This does not change the conclusion that NO$_3$ reactions contribute substantially not only at night-time but also to daytime AN formation in this environment.

Ideally, when comparing the relative contributions of day- and night-time processes to AN formation, we also need to consider the relative volumes of air throughout which the chemistry is taking place. A very rough estimation of how variations in the boundary-layer height can impact on the results can be gained by scaling the alkyl nitrate production rates by the height of the fully developed boundary layer during IBAIRN which was 570 m during the day, and 34 m during the night (Hellén et al., 2018) and integrating over the duration of the day (13h) day or the night (11 h). Inclusion of a factor $\approx$

17 deeper boundary layer during the day results in drastic shift in the relative roles of NO$_3$ versus OH and O$_3$, with only 6% of the total, boundary layer alkyl nitrate production is initiated by NO$_3$ reactions at night-time compared to 39% during the day. By comparison, the daytime, OH initiated formation of alkyl nitrates accounts for 33%, whereas O$_3$ accounts for 22% of the total. These very rough estimates can only be considered illustrative of the potential effects as they are biased by the inherent assumption that there are no vertical gradients in OH or NO$_3$ reactivity during day or night, which is not the case

(Eerdekens et al., 2009; Nölscher et al., 2012; Liebmann et al., 2018a; Zha et al., 2018) and also assumes that the daytime boundary layer reaches its fully developed height immediately after dawn. The true values will depend on the details of mixing within and development of the boundary layer and will lie between the two sets of calculations.

## 3.5 Lifetime of ANs

As illustrated in Fig. 3, the ΣANs mixing ratio reached maximum values of only ~ 100 pptv despite production terms of 40

25   pptv h$^{-1}$, indicating that the lifetimes of the ANs are relatively short. If this is the case, the local concentration of ANs are largely decoupled from the effects of long range transport (timescales of days) and the AN-lifetime ($\tau_{ANs}$) can be can be calculated using a steady-state approach via (Eq. 11):

$$\tau_{ANs} = \frac{[ANs]}{\sum P_{ANs}}$$
(11)

where $\sum P_{ANs}$ is the total production rate (NO$_3$, OH and O$_3$ initiated). The campaign averaged, diel variation of the ANs

mixing ratio and $\sum P_{ANs}$ are given in Fig. 6 (upper panel). A plot of $\Sigma$ ANs versus $\sum P_{ANs}$ using the 1 h averages from the diel profiles is displayed in Fig. 6b with daytime data represented by the red data points and night-time data by the black data points. Within the overall uncertainty represented by the error-bars, there is no significant difference between the day- and

night-time data, with a linear fit through all the data indicating a lifetime of $\approx 2 \pm 3$ hours or a loss rate constant of $5.6 \times 10^{-4}$ $s^{-1}$. The intercept of ~24 pptv ANs can be attributed to small, possibly mono-functional alkyl nitrates that have longer lifetimes and which therefore could have been transported to the site rather than generated locally (Clemitshaw et al., 1997; Romer et al., 2016).

We now examine the potential reasons for the short lifetime of ANs at this location. ANs are generally thought to react inefficiently with $O_3$, OH and $NO_3$ and low rates of photolysis mean that their lifetimes are likely to be controlled largely by dry deposition and / or heterogeneous hydrolysis on aerosol or hydrometeors (Browne et al., 2013). Known exceptions are some ANs formed from isoprene, which can react with OH and/or be photolysed with lifetimes on the order of an hour (Muller et al., 2014; Xiong et al., 2016). During IBAIRN, isoprene derived ANs were however only a small fraction of the

total and, in the absence of kinetic / photochemical data for terpenes, we disregard gas-phase, chemical loss processes.

In a well-mixed daytime boundary layer the deposition velocity ($V_{dep}$) is equal to $k_{dep}H_{BL}$, where $H_{BL}$ is the boundary layer height, and $k_{dep}$ is the first-order loss rate constant due to deposition. Taking a value of $V_{dep}$ ~ 1-2 cm $s^{-1}$ for ANs (Farmer and Cohen, 2008; Nguyen et al., 2015) and an average, noon-time boundary layer height of 570 m, we derive an average lifetime w.r.t. deposition of 8-16 hours, substantially longer than that observed. Conversely, a deposition velocity of ~ 8 cm $s^{-1}$

would result in a lifetime of ~2 hours, consistent with our observations.

The contribution of loss of ANs via heterogeneous uptake to sub-micron aerosol ($k_{het}$) can be assessed via eq. 12.

$$k_{het} = \frac{\gamma \, \bar{c} \, A}{4} \tag{12}$$

Where $\gamma$ is the uptake coefficient, $A$ the aerosol surface area density (in $cm^2$ $cm^{-3}$), $\bar{c}$ the average thermal velocity (in cm $s^{-1}$). The mean aerosol surface area observed during IBAIRN was $2 \times 10^{-7}$ $cm^2$ $cm^{-3}$ (range $0.4 - 6 \times 10^{-7}$ $cm^2$ $cm^{-3}$, see Fig. S3 of

the supplementary information). For a typical C10 alkyl nitrate derived from monoterpene oxidation such as $C_{10}H_{14}NO_7$, (Yan et al., 2016; Lee et al., 2018) $\bar{c}$ ~15000 cm $s^{-1}$ at 290 K. The average uptake coefficient required to reproduce a loss rate constant for ANs of $5.6 \times 10^{-4}$ $s^{-1}$ would then be 0.8, which is orders of magnitude larger than values of $10^{-3}$-$10^{-4}$ reported for water soluble organics (Wu et al., 2015; Crowley et al., 2018). However, the high molecular weight, biogenically derived ANs in the boreal forest have low vapour pressures and transfer via condensation to existing particles is likely to be

important. In this case transfer to the particle phase may be controlled by diffusion and accommodation and the effective uptake efficiency could be much larger. Once transferred to the particle phase, a short lifetime with respect to hydrolysis to $HNO_3$ will result in permanent (irreversible) loss of the AN from the gas-phase (Browne et al., 2013; Bean and Hildebrandt Ruiz, 2016; Lee et al., 2016; Romer et al., 2016; Lee et al., 2018; Zare et al., 2018) and eventually to deposition (as inorganic nitrate) to the plant surfaces and forest floor. The deposition of nitrate thus represents the last step in a sequence of biological

and photochemical processes that enables, via emission of BVOCs, the trapping by the biosphere of reactive, gas-phase $NO_X$ of largely anthropogenic origin and thus the transfer of nitrogen back to the ecosystem. Especially in nitrogen poor environments (e.g. at high latitudes) this represents an important route for plant and forest fertilization (Fowler et al.; Huang et al., 2019).

If, as suggested, the gas-phase organic nitrates formed via the three pathways above are indeed transferred to the aerosol-phase on short times-scales we would expect to find some correlation between the total production rate and aerosol nitrate content, either as organic nitrate or, following hydrolysis, $HNO_3$. AMS measurements of aerosol composition were available for a ~10 day period during the campaign, and we show a plot of AMS-nitrate versus the total ANs production rate in Figure

7. The AMS-nitrate mass loading ($\mu g \ m^{-3}$) was converted to a mixing ratio using a mass of 63 amu (i.e. assuming $HNO_3$).

Figure 7 illustrates that the highest nitrate aerosol content is correlated with high AN production rates, with the ~zero intercept indicating that the formation of particulate nitrate independent of ANs formation is negligible. As the formation of ANs requires $NO_X$, this is not surprising as in the absence of $NO_X$, particulate nitrate formation would also tend to zero. However, a plot of AMS-nitrate versus $NO_X$ over the same period (Fig S4 of the supporting information) is more scattered,

supporting the contention that the combination of BVOC oxidation in the presence of $NO_X$ (i.e. ANs formation) is a major source of aerosol nitrate in this environment.

By taking up ANs, particles effectively integrate the ANs production term over time and the slopes of the solid lines in Fig. 7 represent integration times of 2.8 h (upper bound) and 0.5 h (lower bound) factored by the efficiency of uptake. The latter may be related to several factors that control the transfer of gas-phase ANs to the particle phase, including relative humidity,

temperature, available aerosol surface area, release of $HNO_3$ back to the gas-phase and (competitively) dry-deposition. Colour coding the data in Fig. 7 for various parameters (temperature, relative humidity, aerosol surface area or other particle properties (ammonium, sulphate, organic mass) revealed that that the larger slopes are associated with higher organic content (Fig. S4), which in turn is expected to be associated with more aged aerosol. No trend was found in parameters such as temperature and relative humidity, suggesting that their influence on the transfer of ANs to the particle phase is weak. We

cannot explore the role of dry-deposition of ANs in detail, but suggest that this is unlikely to vary sufficiently to induce the observed variation in the slopes observed in Fig. 7. If we assume 100 % transfer of ANs to the particle phase, the integration time (upper bound to the data in Fig. 7) represents the maximum lifetime (with respect to deposition) of 2.8 hrs for the aerosol.

Our TD-CRDS instrument measures the total concentration of ANs, without speciation and therefore does not allow us to

investigate whether ANs formed via OH-, $O_3$- or $NO_3$-initiated degradation of BVOCs have different lifetimes. The molecular composition of a total of 45 organic nitrates (C1-C10, O3-O8, N1) was identified using the HR-L-ToF-CIMS, which is known to be sensitive towards alkyl-nitrates (Lee et al., 2014b; Lee et al., 2016). In the following discussion we therefore consider the masses to be associated mainly with alkyl-nitrates (and not e.g. peroxy-nitrates). Neither the absolute nor the relative sensitivity (between different alkyl-nitrates) is known and the following discussion is necessarily qualitative

in nature. Assuming equal sensitivity across the mass spectrum for the HR-L-ToF-CIMS measurements in IBAIRN (Lee et al., 2014a; Lee et al., 2016) we find that in total only 10% of the organic nitrates were C5, whereas C6-C10 accounted for 85% of the signal. This is indicated in Fig. S4 of the supplementary information.

In Fig. 7, we compare the TD-CRDS measurement of the ΣANs measurement to signal of the HR-L-ToF-CIMS, the latter separated into C1-C5 and C6-C10 species. As the HR-L-ToF-CIMS data are available as counts only, we have normalised

the datasets in Figure 6 to the values at 12:00 UTC. The C1-C5 and C6-C10 trace gases identified by the HR-L-ToF-CIMS display very similar diel cycles, with daytime maxima and low values at night-time, as also seen for the ΣANs data. The HR-L-ToF-CIMS data do however indicate a much more pronounced diel cycle, which is related to the different positions of the inlets of the instruments. Whereas the TD-CRDS sampled at a height of 8.5 m above the ground, the HR-L-ToF-CIMS inlet was located at a height of just 1.5 m. As mentioned already, high relative humidity was frequently accompanied by ground-level fog at night-time during IBAIRN, which would have significantly impacted the HR-L-ToF-CIMS dataset, resulting in lower, mean night-time signals. In addition, we cannot rule out that this difference is due to different HR-L-ToF-CIMS sensitivity to day- and night-time ANs.

Figure 8 plots the daytime and night-time campaign mean signal at each organic nitrate mass throughout the campaign. C9 and C10 nitrates provided the greatest signals on average, followed by C7 and C8. As seen from the diel average (Fig. 7), the organic-nitrate signals were significantly larger during day than at night-time, the lowest day-to-night ratios being found among the C9 and C10 species and the largest day-night ratios found for C5 and smaller. The C5 organic nitrates (and smaller) are likely to stem from isoprene chemistry, whereas C6 and larger are likely to be products of terpene oxidation (Lee et al., 2016). Our observations are broadly consistent with the study of Huang et al. (2019) in a mixed isoprene-terpene forest, who observed daytime concentration maxima for C5-species and night-time maxima for C10. A more detailed comparison with Huang et al. (2019) is however difficult owing to very different isoprene-to-terpene emission rates, which in the IBAIRN campaign strongly favoured terpenes (Liebmann et al., 2018a).

## 4 Conclusions

During the IBAIRN campaign in the boreal forest in southern Finland (5[th]-22[nd] Sept, 2016), alkyl nitrate formation was dominated by the reaction of $NO_3$ radicals with monoterpenes, both during the day- and night-time, with smaller contributions from both OH and $O_3$ initiated oxidation of BVOCs. This result highlights the important role of daytime $NO_3$ chemistry (with respect to organic nitrate formation) in this environment. The short, average lifetime of $\approx$ 2 h for the total alkyl-nitrates (ΣANs) indicates efficient uptake to existing particles and/or deposition.

These observations, of efficient daytime production of gas-phase ANs from $NO_3$ chemistry and short night-time lifetimes are entirely consistent with the results from recent studies at the IBAIRN site by Lee et al. (2018) who found that organic nitrates previously designated as resulting from night-time processing of BVOCS (Yan et al., 2016) were also present during daytime. In addition, they found relatively few organics with "night-time" character in the gas-phase compared to the aerosol-phase, indicating efficient transfer of gas-phase organic nitrates to the particle-phase at night-time, likely aided by low temperatures and high relative humidity. We found no significant change in the production rate of ANs in transition from summer to autumn, though the short duration of the campaign and variability in temperature and insolation would mask such effects.

## Data availability

The MPI-data used in this study are archived with Zenodo under the following DOI: 10.5281/zenodo.3254828. Contingent to agreeing with the IBAIRN data-protocol, the data will be available for external users from August 2019.

## Author contributions

Jonathan Liebmann was responsible for the $NO_3$-reactivity measurements and interpretation during IBAIRN and, with contributions from John Crowley, Horst Fischer and Jos Lelieveld, wrote the manuscript. Nicolas Sobanski was responsible for the CRDS measurements of ANs, $NO_3$ and $NO_2$. Jan Schuladen was responsible for the $O_3$ and J-value measurements. Horst Fischer was responsible for the NO measurements. Einar Karu, Jonathan Williams Heidi Hellén and Hannele Hakola were responsible for the VOCs and BVOCs measurements. Qiaozhi Zha and Matthieu Riva were responsible for the I-CIMS measurements of ANs. The IBAIRN campaign was conceived and organised by John Crowley and Mikael Ehn.

## Competing interests

The authors declare that they have no conflict of interest.

## Acknowledgements

We are grateful to ENVRIplus for partial financial support of the IBAIRN campaign. We would like to thank Uwe Parchatka for the provision of NO measurements and the excellent technical support from Janne Levula and team at Hyytiälä.

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

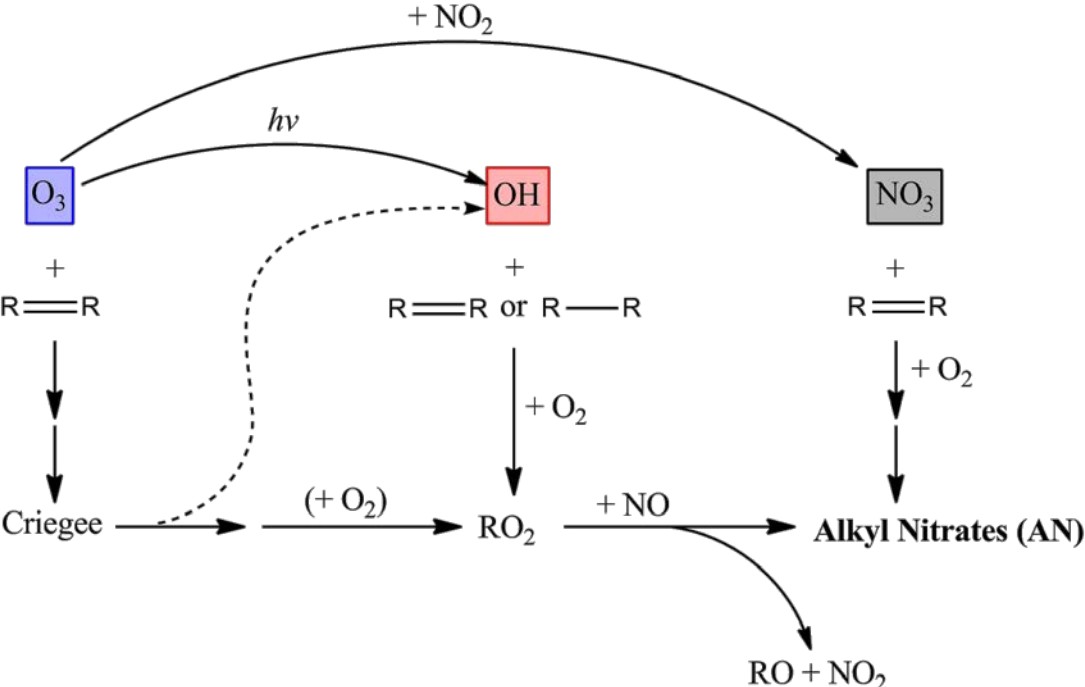

**Figure 1.** Schematic diagram illustrating the formation of ANs via VOC degradation initiated by $NO_3$, OH and $O_3$ reactions.

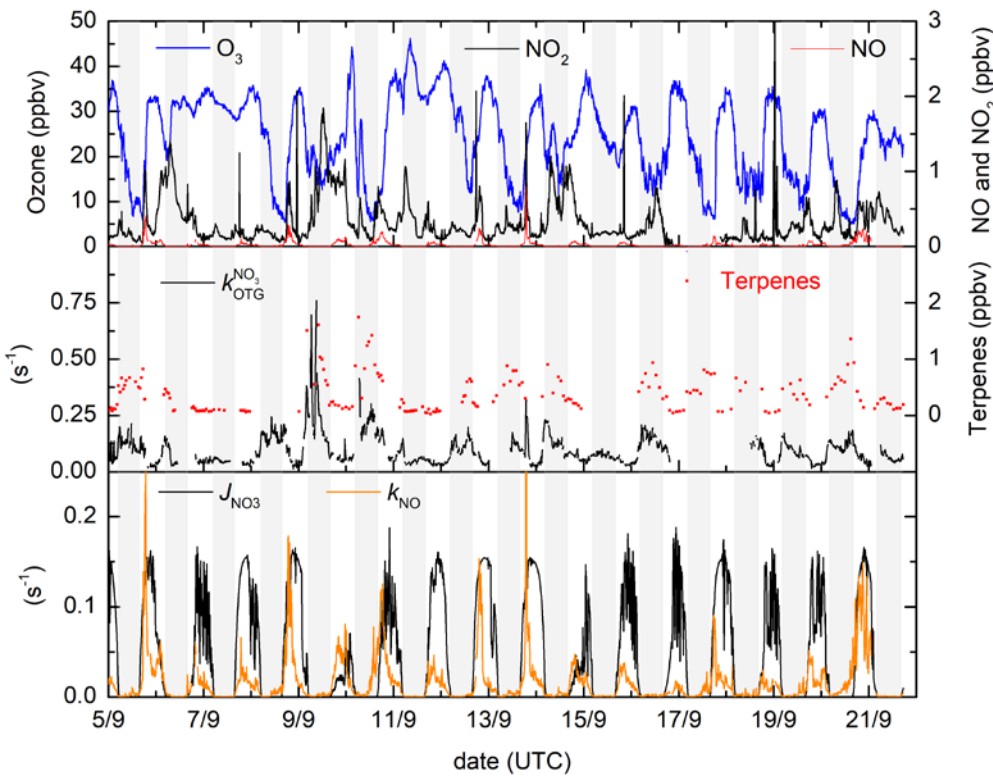

**Figure 2.** Time series of the NO$_3$ precursors (NO$_2$ and O$_3$), the NO$_3$ reactivity to organic trace gases ($k_{OTG}^{NO_3}$) and the first-order loss-constants for its photolysis ($J_{NO3}$) and reaction with NO ($k_{NO}$) during IBAIRN. Grey shaded regions are night-time.

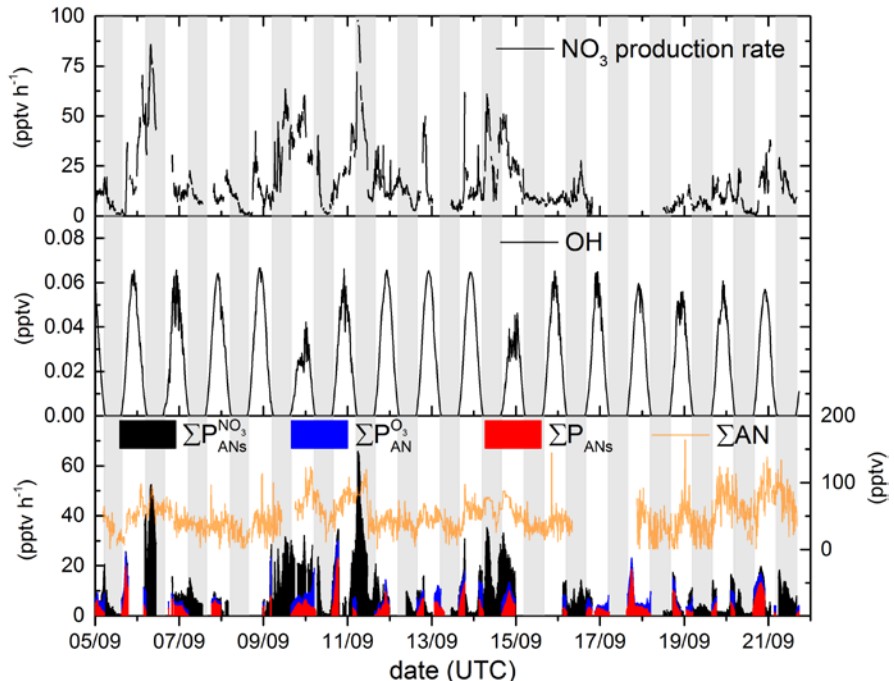

**Figure 3.** Upper Panel: $NO_3$ radical production rate from reaction of $NO_2$ and $O_3$. Middle Panel: OH mixing ratio as derived from Eq. 7. Lower Panel: Production rate of ANs from OH (red), $O_3$ (blue) and $NO_3$ radicals (black) and the $\Sigma AN$ mixing ratio (orange line).

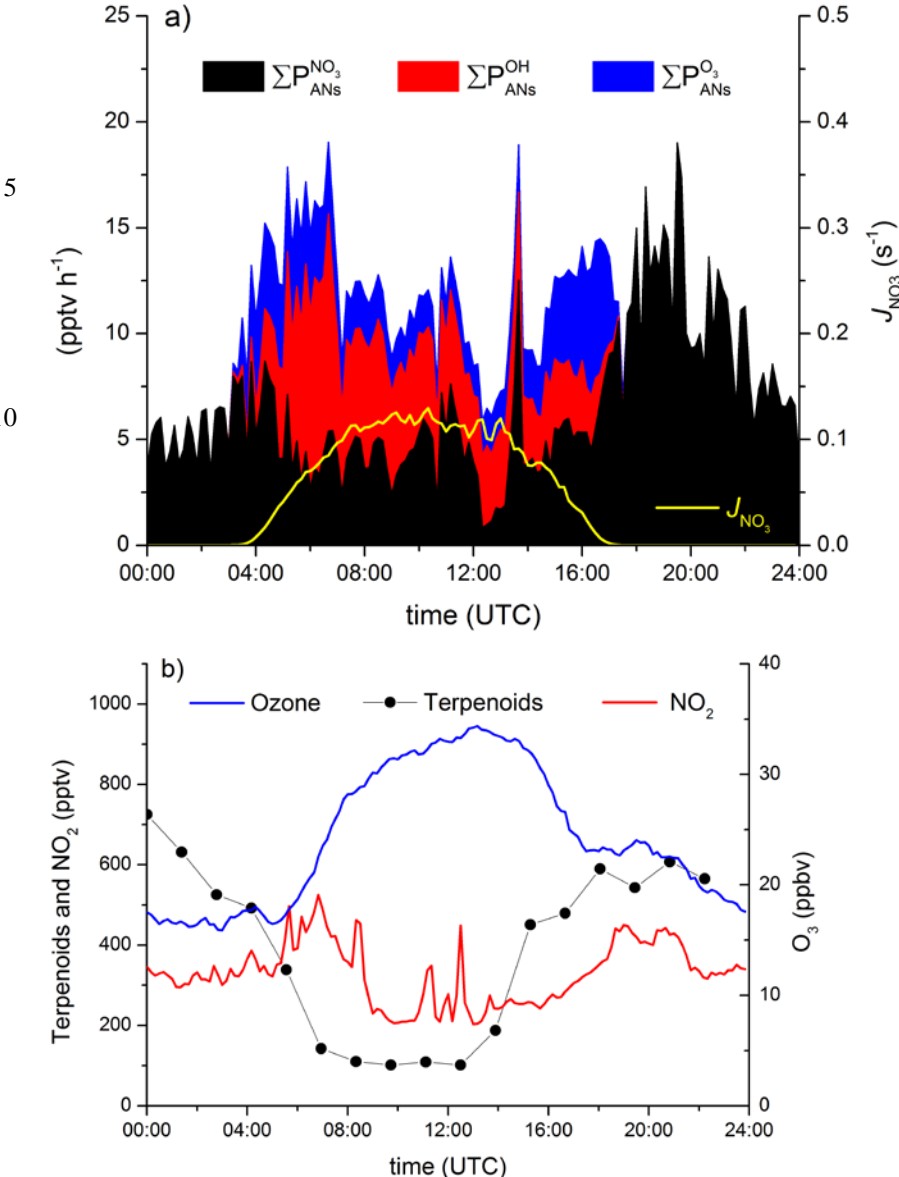

**Figure 4.** a) Diel profiles (means) of the AN production rates for VOC initiated oxidation by the OH radical ($\Sigma P_{AN}^{OH}$), $O_3$ ($\Sigma P_{AN}^{O3}$) and the $NO_3$ radicals ($\Sigma P_{AN}^{NO3}$) and the photolysis rate constant of $NO_3$ (yellow line) to distinguish in between night and day. b) Diel profiles (means) of the mixing ratios of $O_3$, $NO_2$ and total terpenoids.

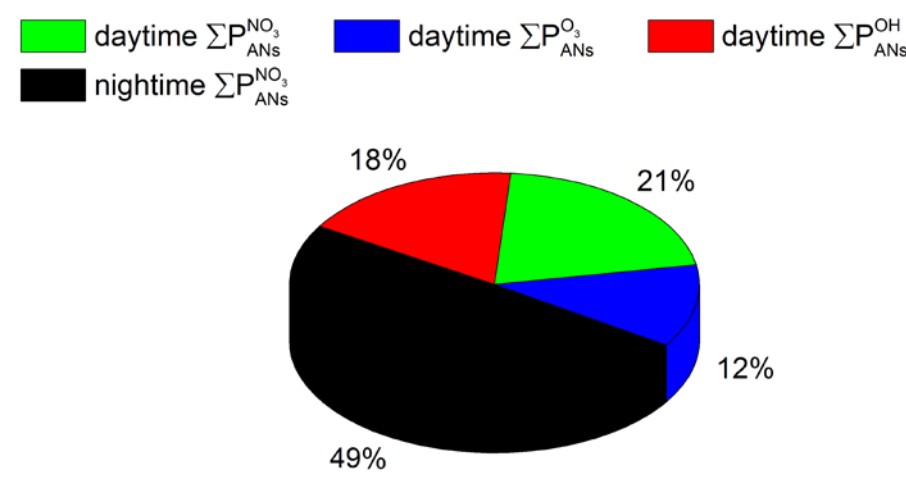

**Figure 5.** Contribution of NO₃-, OH-, and O₃ – initiated degradation of VOCs to the overall formation of alkyl nitrates.

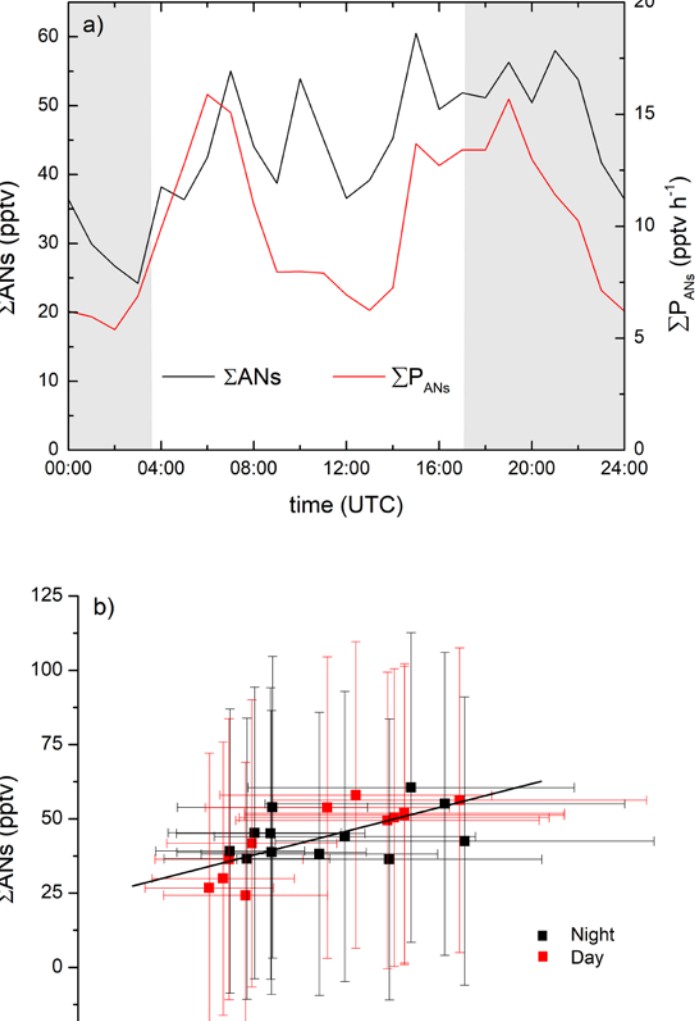

**Figure 6**: Upper panel: Diel profiles of production rate and mixing ratios of ANs during IBAIRN. Lower panel: Plot of ΣAN mixing ratios versus the total production rate from $NO_3$-, OH- and $O_3$-initiated VOC oxidation during day (06:00-15:00 UTC, black dots) and night-time (18:00-03:00 UTC, red dots). The slope of the linear fit to the data (York-fit, errors in both axes considered, black line) indicates a lifetime of 2±3 h. The error bars correspond to the total uncertainty as described in the text.

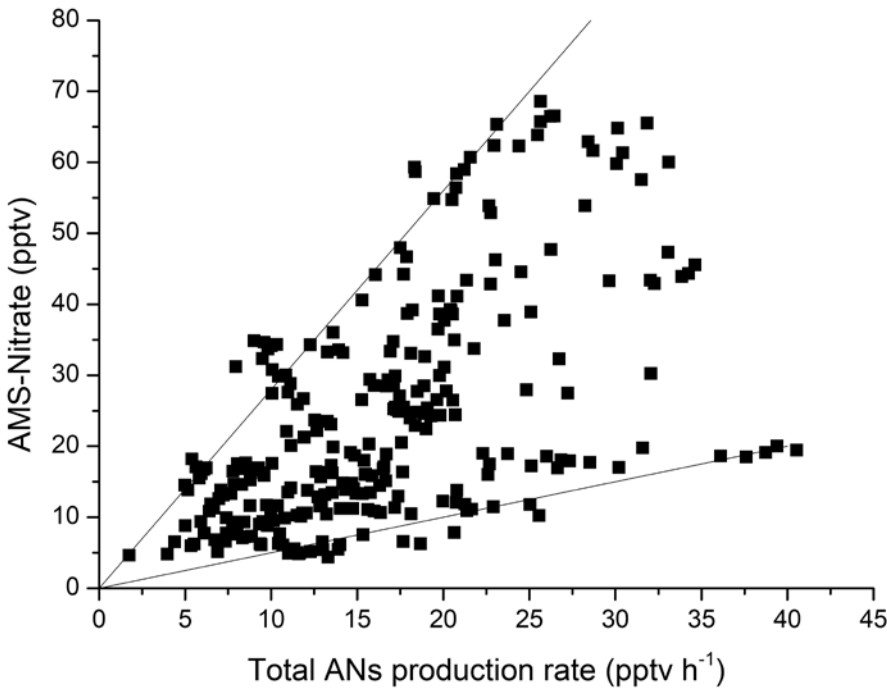

**Figure 7.** Total particle nitrate (AMS) versus the total production rate of ANs from reactions of $NO_3$, OH and $O_3$ with BVOCs. The solid lines are upper and lower bounds with slopes of ~ 2.8 hours, and 0.5 hours, respectively.

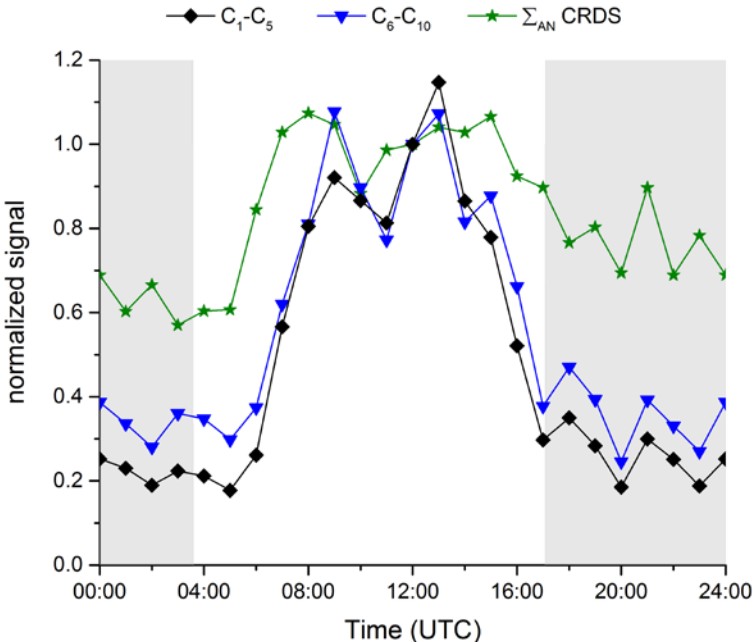

**Figure 8.** Relative diel profile of I-CIMS signals attributed to C1-C5 and C6-C10 organic nitrates and comparison with ΣANs measured by the TD-CRDS. Data normalized to 1 at 12:00 UTC.

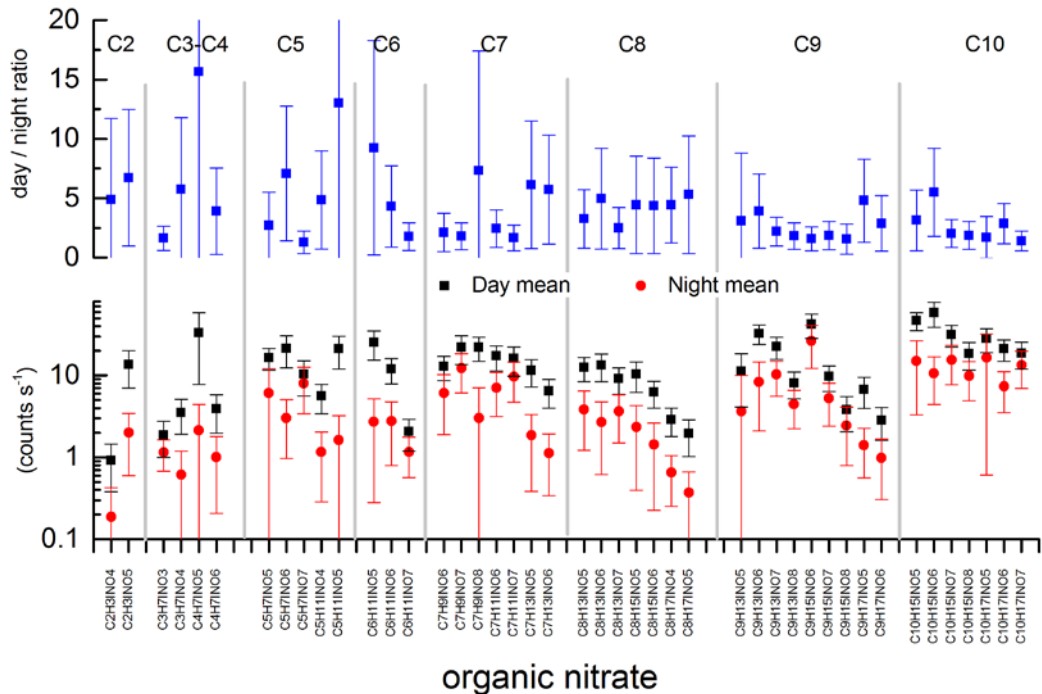

**Figure 9.** Campaign mean signals for organic nitrates measured by the I-CIMS. The I-CIMS was not calibrated thus only the raw signal (counts per second) at each mass is given.

**Figure Captions:**

**Figure 1.** Schematic diagram illustrating the formation of ANs via VOC degradation initiated by $NO_3$, OH and $O_3$ reactions.

**Figure 2.** Time series of the $NO_3$ precursors ($NO_2$ and $O_3$), the $NO_3$ reactivity to organic trace gases ($k_{OTG}^{NO_3}$) and the first-order loss-constants for its photolysis ($J_{NO3}$) and reaction with NO ($k_{NO}$) during IBAIRN. Grey shaded regions are night-time.

**Figure 3.** Upper Panel: $NO_3$ radical production rate from reaction of $NO_2$ and $O_3$. Middle Panel: OH mixing ratio as derived
from Eq. 7. Lower Panel: Production rate of ANs from OH (red), $O_3$ (blue) and $NO_3$ radicals (black) and the $\Sigma$AN mixing ratio (orange line).

**Figure 4.** a) Diel profiles (means) of the AN production rates for VOC initiated oxidation by the OH radical ($\Sigma P_{AN}^{OH}$), $O_3$ ($\Sigma P_{AN}^{O3}$) and the $NO_3$ radicals ($\Sigma P_{AN}^{NO3}$) and the photolysis rate constant of $NO_3$ (yellow line) to distinguish in between night
and day. b) Diel profiles (means) of the mixing ratios of $O_3$, $NO_2$ and total terpenoids.

**Figure 5.** Contribution of $NO_3$-, OH-, and $O_3$ – initiated degradation of VOCs to the overall formation of alkyl nitrates.

**Figure 6**: Upper panel: Diel profiles of production rate and mixing ratios of ANs during IBAIRN. Lower panel: Plot of $\Sigma$AN
mixing ratios versus the total production rate from $NO_3$-, OH- and $O_3$-initiated VOC oxidation during day (06:00-15:00 UTC, black dots) and night-time (18:00-03:00 UTC, red dots). The slope of the linear fit to the data (York-fit, errors in both axes considered, black line) indicates a lifetime of $2\pm3$ h. The error bars correspond to the total uncertainty as described in the text.

**Figure 7.** Total particle nitrate (AMS) versus the total production rate of ANs from reactions of $NO_3$, OH and $O_3$ with BVOCs. The solid lines are upper and lower bounds with slopes of ~ 2.8 hours, and 0.5 hours, respectively.

**Figure 8.** Relative diel profile of I-CIMS signals attributed to C1-C5 and C6-C10 organic nitrates and comparison with $\Sigma$ANs measured by the TD-CRDS. Data normalized to 1 at 12:00 UTC.

**Figure 9.** Campaign mean signals for organic nitrates measured by the I-CIMS. The I-CIMS was not calibrated thus only the raw signal (counts per second) at each mass is given.