# Peer review of "Alkyl nitrates in the boreal forest: Formation via the NO3, OH and O3 induced oxidation of BVOCs and ambient lifetimes"

_Atmospheric Chemistry and Physics, 2019_

## Referee Comment (RC1) · Anonymous Referee #1 · 5 Jun 2019

Review of Liebmann et al., 2019: "Alkyl nitrates in the boreal forest: Formation via the NO3, OH, and O3 induced oxidation of BVOCs and ambient lifetimes"

This manuscript uses measurements from a boreal forest to compare the relative importance of alkyl nitrate formation via NO3, O3, and OH oxidation pathways during both day and night. They find, somewhat surprisingly, that NO3 oxidation of BVOCs accounts for up to half the daytime production of alkyl nitrates, and that there are approximately equal rates of alkyl nitrate production during day and during night. Additionally, the authors calculate a relatively short steady-state alkyl nitrate lifetime of 2 hours, implying that heterogeneous hydrolysis is likely an important loss process in this

environment.

These are interesting results on the fractional contribution of different oxidation pathways to alkyl nitrate production and on the lifetime of alkyl nitrates in a boreal forest. The paper is well organized and does a nice job accounting for the uncertainties in various calculations. I recommend publication.

Some questions and comments to improve the manuscript can be found below.

Major comments: 1. Some parts of the manuscript would benefit from being more quantitative. For example, on page 5, line 14 (and similarly on page 7, line 1), the authors state that "only the handful of biogenic VOCs listed contributed significantly." More quantification (i.e., biogenic VOCs contributed to >x% of the observed reactivity) would be helpful. Additionally, (page 10, line 6), please be quantitative and specify the aerosol surface area measured, instead of discussing it in purely relative terms.

2. It would be helpful if the authors were able to observationally constrain some of the numbers they estimate in the manuscript. For example, on page 5, line 26 the authors assign an alkyl nitrate yield of 0.7 to unattributed VOCs because they suspect the missing reactivity is from highly reactive BVOCs with high yields. Could this number be observationally constrained using the quantified missing reactivity and the ANs production rate? Likewise, is it possible to get an observational constraint on the alkyl nitrate deposition velocity (page 9, line 29)? Or at least compare the HNO3 production implied by the estimated hydrolysis rates to observed increases in aerosol inorganic nitrate? And lastly (page 10), can you use the aerosol surface area that was measured to estimate an aerosol uptake efficiency for alkyl nitrates, rather than simply saying the "efficiency could be >0.1"?

3. I am curious how much the seasonal changes over the course of the IBAIRN study affected the various production and loss processes for alkyl nitrates. Do averages from the first half of the study (summer) and the second half of the study (autumn) give significantly different results, or are there minimal differences?

[Figure]

4. Is it possible to connect the individual alkyl nitrates observed by CIMS to the ANs production rates of alkyl nitrates calculated from individual VOCs? Could this give any indication as to which VOCs contribute to the missing reactivity?

Minor comments: 1. Page 2, line 12: Why is reaction with $O_3$ only relevant in the boreal forest? 2. Page 2, line 30: Clarify to say "...the branching ratio to AN formation via $NO_3$ oxidation is generally..." 3. Equation 4: Typo–should include $k5$ rather than $k3$. 4. Page 6: Should $RO_2$ loss via reaction with $NO_2$ to form PANs also be accounted for? Or is it insignificant? 5. Page 8, line 8: Clarify to say that the ANs production from ozonolysis has a daytime minimum at noon (since the absolute minimum is really at night). 6. Figures 2 and 3: x-axis labels are confusing. 7. Page 9, line 15: I think your estimate uses the "steady-state" approximation rather than the "stationary-state" approximation. 8. Figure 6: Please define what your error bars are (standard deviation?). Additionally, the ends of some of the error bars are not visible in the plot. Is the fit you are doing to all points or only the average points that are plotted? What kind of fit are you using (OLS, RMA, York?)? 9. Page 9, line 25: Some alkyl nitrates (e.g., isoprene hydroxy nitrate) can be oxidized by OH with reasonable efficiency, and highly oxidized or carbonyl nitrates can be rapidly photolyzed (see Muller et al., 2014 and Xiong et al., 2016). Are these not relevant during IBAIRN? 10. Page 9, line 32: Should be "assessed" instead of "accessed." 11. Page 10, line 3: Was the calculation of average thermal velocity done at STP or at the average temperature and pressure during the campaign? Please specify. 12. Page 10, line 11: Consider citing Romer et al., 2016 and Zare et al., 2018 which also discuss ANs lifetimes and heterogeneous hydrolysis as a loss pathway for ANs. 13. A separate conclusion section would be helpful to the reader (i.e. add a section header before the last two paragraphs).

---

## Referee Comment (RC2) · Anonymous Referee #2 · 24 Jun 2019

**Review of Liebmann et al., "Alkyl nitrates in the boreal forest: Formation via the NO3, OH and O3 induced oxidation of BVOCs and ambient lifetimes," for ACP May 2019**

This concise and clear paper reports on measurements of alkyl nitrates in the boreal forest, with coincident measurements of organic trace gases enabling an assessment of the relative source strength of OH, NO3, and O3 oxidation in producing these alkyl nitrates. In this NOx-limited environment, NO3 oxidation is found to be the dominant source of alkyl nitrates both night and day. The paper is clearly written and the figures are helpful. I suggest addition of a bit more auxiliary data to enable readers to better interpret the conclusions.

**General comments:**

1) As I read this paper and sought to understand the key observations, I found myself wondering about the [NO] and relative concentrations of different organic trace gases. These data are perhaps in other papers cited, but for convenience of the reader I urge the authors to include this data here. I suggest to include an NO trace in the top panel of Figure 2, and add a panel to that figure showing BVOC timeseries, perhaps split out by isoprene and summed terpenes since they likely have different diel patterns, and since their relative reactivities is different and can help the reader interpret the day/night observations.

2) For similar reasons, it would be helpful to add another panel to the diel average figure 4, showing NO2 and BVOC traces. In particular, I was curious why the NO3-initiated production of Ans would peak at 19:00 local time and then decrease? Given your statement that monoterpene concentrations build up overnight, I might have expected this to continue increasing. Is it that the NO2 is fully consumed by then?

3) I agree with Jacqui, Figure 5 is not necessary.

4) Abstract line 22: "strongly controlled by biogenic emissions" – ? seems inconsistent with your discussion in the manuscript body. There, you describe this as due to rapid deposition to particles?

5) Around p. 3 line 23-24: Could you list the dominant terpenes here? (and / or, on p. 5 around line 13 where you state that only a handful contributed significantly to reactivity – include a brief ranked list?) Also, what anthropogenic emissions are observed from the cities & sawmill – just NOx, or NOx and SO2?

6) On p. 8 around line 18: say something about why the NO3 initiated formation of ANs peaks at 19:00

7) On Figure 6, the error bars on the bottom panel look very large compared to the reported slope uncertainty of +/- 0.5 hr. Please explain how this error bar is determined – it looks to me like the slope could even be negative within the uncertainties.

8) Figure 7 makes me wonder whether it's possible that different sensitivities of I- CIMs to daytime vs. nighttime BVOC mixes could explain the different amplitude of the diel cycle. Can you add anything additional information on this?

**Technical corrections/suggestions:**

p. 2 line 21: "OH radicals are largely absent"

line 27: "Reaction 6a is a composite"

p. 3 line 10: suggest to add reference to Ayres 2015 (https://www.atmos-chem-phys.net/15/13377/2015/): this NO3 + BVOC dominance during the day was also observed at SOAS 2013.

p. 3 line 30 "reached 100% during many nights"

end p. 3 / top of p. 4: Is 300 pptv the average NOx level for the whole campaign? Maybe also mention the [NOx] during the events where airmasses arrive from the industrial sources.

p. 4 line 4 "photolysis frequency, and the"

line 21: remove extra ")"

lines 25-26: "OH concentrations have an associated uncertainty of ~ 50%"

line 33: add citation for I- CIMS high sensitivity to nitrates

p. 5 Eq. 2: It's a little confusing that you use the average alpha in the equation but then talk about the individual ones first below the equation, and then define the average in Eq 3 below. Maybe combine Eq. 3 into 2 so you see the average and the summation simultaneously? Also, after the current Eq. 3, define the Ci term.

Eqn 4: k3 should be k5?

Line 24 "total measured reactivity"

p. 6 Eqn R12: meaning of the "delta" term is unclear

line 27: "UTC). In order to account for this competition with HO2 reactions, equation (7) can be modified to:"

p. 7 line 3: "the local sawmill, likely due to elevated reactivity with ….?"

Line 4-5: This sentence sounds like you're drawing a contrast to NO3, but I think this is true in that case as well. Perhaps make this sentence the first sentence of the next paragraph instead?

Line 11: "We show below that even if unattributed OH-reactivity reaches 50%, this would not significantly"

Line 13: insert space "from [Ci]"

Line 20: "Similarly as for OH-reactions,"

p. 8 line 2: "was estimated to be ~ 60%"

line 10: "is rather consistent with"

line 24: "ANs production rate occurs exclusively via NO3-initiated reactions."

p. 9 line 1 "which was 570 m"

line 2: omit extra ")"

line 7: "which is not the case (Eerdekens"

line 20: "overall uncertainty represented by error bars, there"

line 27: "well-mixed daytime boundary"

line 28: include units on Vdep (here and in the line below "Vdep ~ 2")

line 32: "can be assessed"

p. 10 line 3: "For typical alkyl nitrate"

line 18 & below: Shouldn't the CIMS be designated the "I- CIMS" and not "I-CIMS"?

p. 11 top 2 lines: C9 shows up in two categories ?

line 11: "lifetimes are entirely consistent"

---

## Author Comment (AC1) · 12 Jul 2019

**Referee 1**

In the following, the referee's comments are reproduced (black) along with our replies (blue) and changes made to the text (red) in the revised manuscript.

**General Comments:**

This manuscript uses measurements from a boreal forest to compare the relative importance of alkyl nitrate formation via $NO_3$, $O_3$, and OH oxidation pathways during both day and night. They find, somewhat surprisingly, that $NO_3$ oxidation of BVOCs accounts for up to half the daytime production of alkyl nitrates, and that there are approximately equal rates of alkyl nitrate production during day and during night. Additionally, the authors calculate a relatively short steady-state alkyl nitrate lifetime of 2 hours, implying that heterogeneous hydrolysis is likely an important loss process in this environment. These are interesting results on the fractional contribution of different oxidation pathways to alkyl nitrate production and on the lifetime of alkyl nitrates in a boreal forest. The paper is well organized and does a nice job accounting for the uncertainties in various calculations. I recommend publication. Some questions and comments to improve the manuscript can be found below.

We thank the referee for this positive assessment of our manuscript.
* * *
**Major comments:**

1. Some parts of the manuscript would benefit from being more quantitative. For example, on page 5, line 14 (and similarly on page 7, line 1), the authors state that "only the handful of biogenic VOCs listed contributed significantly." More quantification (i.e., biogenic VOCs contributed to >x% of the observed reactivity) would be helpful.

We now write:

(P5L14) A large selection of VOCs was measured (a listing is given in the caption to Figure S2 of the supplementary information) but the 5 biogenic VOCs listed ($\alpha$-pinene, $\beta$-pinene, carene, limonene, isoprene) accounted for > 98 % of the attributed $NO_3$ reactivity.

(P7L1) The total OH-reactivity ($k_{OTG}^{OH}$) was calculated using the measured concentrations of monoterpenes and isoprene as other VOCs (e.g. aldehydes, acids, alkanes, alkenes) contributed less than 6% (Fig S2).
* * *
Additionally, (page 10, line 6), please be quantitative and specify the aerosol surface area measured, instead of discussing it in purely relative terms.

Done: (see also comment below about the uptake coefficient)

The aerosol surface area is $NO_3$ plotted in (additional) Fig. S3 of the supplementary information.
* * *
2. It would be helpful if the authors were able to observationally constrain some of the numbers they estimate in the manuscript. For example, on page 5, line 26 the authors assign an alkyl nitrate yield of 0.7 to unattributed VOCs because they suspect the missing reactivity is from highly reactive BVOCs with high yields. Could this number be observationally constrained using the quantified missing reactivity and the ANs production rate?

The uncertainty associated with the calculation of the ANs production rate (see sections 3.1, 3.2 and 3.3) are large (65% for $NO_3$, 70% for OH and 60% from $O_3$) and can therefore not be usefully applied to constrain e.g. the yield of ANs from unattributed reactivity of $NO_3$.
* * *
Likewise, is it possible to get an observational constraint on the alkyl nitrate deposition velocity (page 9, line 29)?

We cannot constrain the deposition velocity as the loss of ANs is (likely) due to a combination of deposition and particle uptake / hydrolysis. However, considered individually we can place an upper limit: We now write:

Taking a value of $V_{dep} \sim 1\text{-}2$ cm s$^{-1}$ for ANs (Farmer and Cohen, 2008; Nguyen et al., 2015) and an average, noon-time boundary layer height of 570 m, we derive an average lifetime w.r.t. deposition of 8-16 hours, substantially longer than that observed. Conversely, a deposition velocity of $\sim 8$ cm s$^{-1}$ would result in a lifetime of ~2 hours, consistent with our observations.
* * *
Or at least compare the HNO3 production implied by the estimated hydrolysis rates to observed increases in aerosol inorganic nitrate?

The short lifetime of the ANs indicates that a large fraction is transferred to the particle phase, the rest being lost by deposition. As the referee correctly infers, this must result in an increase in particulate nitrate either as organic nitrate or, after hydrolysis, as $HNO_3$. Aerosol composition measurements (AMS) were available for the campaign, though the instrument was operational only for about 10 days altogether between the 5th and 22nd of Sept. We have correlated the AMS-nitrate measurement to the total production rate and added a Figure (new Fig. 7) and text to describe the results.

If, as suggested, the gas-phase organic nitrates formed via the three pathways above are indeed transferred to the aerosol-phase on short times-scales we would expect to find some correlation between the total production rate and aerosol nitrate content, either as organic nitrate or, following hydrolysis, $HNO_3$. AMS measurements of aerosol composition were available for a ~10-day period during the campaign, and we show a plot of AMS-nitrate versus the total ANs production rate in Figure 7. The AMS-nitrate mass loading ($\mu g$ m$^{-3}$) was converted to a mixing ratio using a mass of 63 amu (i.e. assuming $HNO_3$).

The Figure illustrates that the highest nitrate aerosol content is correlated with high AN production rates, with the ~zero intercept indicating that the formation of particulate nitrate independent of ANs formation is

negligible. As the formation of ANs requires $NO_X$, this is not surprising as in the absence of $NO_X$, particulate nitrate formation would also tend to zero. However, a plot of AMS-nitrate versus $NO_X$ over the same period (Fig S4 of the supporting information) is more scattered, supporting the contention that the combination of BVOC oxidation in the presence of $NO_X$ (i.e. ANs formation) is a major source of aerosol nitrate in this environment.

By taking up ANs, particles effectively integrate the ANs production term over time and the slopes of the solid lines in Fig. 7 represent integration times of 2.8 h (upper bound) and 0.5 h (lower bound) factored by the efficiency of uptake. The latter may be related to several factors that control the transfer of gas-phase ANs to the particle phase, including relative humidity, temperature, available aerosol surface area, release of $HNO_3$ back to the gas-phase and (competitively) dry-deposition. Colour-coding the data in Fig. 7 for various parameters (temperature, relative humidity, aerosol surface area or other particle properties (ammonium, sulphate, organic mass) revealed that that the larger slopes are associated with higher organic content (Fig. S4), which in turn is expected to be associated with more aged aerosol. No trend was found in parameters such as temperature and relative humidity, suggesting that their influence on the transfer of ANs to the particle phase is weak. We cannot explore the role of dry-deposition of ANs in detail, but suggest that this is unlikely to vary sufficiently to induce the observed variation in the slopes observed in Fig. 7. If we assume 100 % transfer of ANs to the particle phase, the integration time (upper bound to the data in Fig. 7) represents a maximum lifetime (with respect to deposition) of 2.8 hrs for the aerosol.
* * *
And lastly (page 10), can you use the aerosol surface area that was measured to estimate an aerosol uptake efficiency for alkyl nitrates, rather than simply saying the "efficiency could be >0.1"?

We have made this calculation and added an extra figure (aerosol surface area) to the supplementary information. We now write.

Where $\gamma$ is the uptake coefficient, $A$ the aerosol surface area density (in $cm^2$ $cm^{-3}$), $\bar{c}$ the average thermal velocity (in $cm$ $s^{-1}$). The mean aerosol surface area observed during IBAIRN was $2 \times 10^{-7}$ $cm^2$ $cm^{-3}$ (range $0.4 - 6 \times 10^{-7}$ $cm^2$ $cm^{-3}$, see Fig. S3 of the supplementary information). For a C10 alkyl nitrate derived from monoterpene oxidation such as $C_{10}H_{14}NO_7$, (Yan et al., 2016; Lee et al., 2018) $\bar{c} \sim 15000$ $cm$ $s^{-1}$ at 290 K. The average uptake coefficient required to reproduce a loss rate constant for ANs of $5.6 \times 10^{-4}$ $s^{-1}$ would then be 0.8, which is orders of magnitude larger than values of $10^{-3}$-$10^{-4}$ reported for water soluble organics (Wu et al., 2015; Crowley et al., 2018). However, the high molecular weight, biogenically derived ANs in the boreal forest have low vapour pressures and transfer via condensation to existing particles is likely to be important. In this case transfer to the particle phase may be controlled by diffusion and accommodation and the effective uptake efficiency could be much larger.
* * *
3. I am curious how much the seasonal changes over the course of the IBAIRN study affected the various production and loss processes for alkyl nitrates. Do averages from the first half of the study (summer) and the second half of the study (autumn) give significantly different results, or are there minimal differences?

Within the measurement uncertainty, there is no observable trend in the ANs production (or concentration) when comparing the first and seconds halves of the campaign. Short term variations in temperatures, insolation etc. are too large (conversely, the campaign was too short). We write:

We found no significant change in the production rate of ANs in transition from summer to autumn, though the short duration of the campaign and variability in temperature and insolation would mask such effects.
* * *
4. Is it possible to connect the individual alkyl nitrates observed by CIMS to the production rates of alkyl nitrates calculated from individual VOCs? Could this give any indication as to which VOCs contribute to the missing reactivity?

This is a good idea, but unfortunately, in this case, not feasible. As we state in the text, the I-CIMS data are only used in a qualitative sense owing to the different inlet positions and heights (and diel profiles) when compared to e.g. the $\Sigma$ANs or $NO_3$-reactivity measurements, and are not suitable for investigation of the individual contributions of single BVOC to AN formation.
* * *
**Minor comments:**

1. Page 2, line 12: Why is reaction with $O_3$ only relevant in the boreal forest?

We did not seek to imply this and have made the text more general:

The reaction with $O_3$ represents an additional sink for biogenic VOCs (BVOCs) (Peräkylä et al., 2014; Yan et al., 2016) which, in the presence of NO, can also lead to the formation of alkyl nitrates.
* * *
2. Page 2, line 30: Clarify to say "… the branching ratio to AN formation via NO3 oxidation is generally…"

Correction made. We now write:

"The branching ratio to AN formation via $NO_3$ oxidation is generally much larger than that for organic peroxy radicals reacting with NO…."
* * *
3. Equation 4: Typo–should include k5 rather than k3.

Correction made
* * *
4. Page 6: Should RO2 loss via reaction with NO2 to form PANs also be accounted for? Or is it insignificant?

Only a small fraction of the total $RO_2$ (i.e. $\alpha$-carbonyls) are capable of forming a stable PAN, other $RO_2$ + $NO_2$ interactions are reversible on time scales of minutes and therefore not considered. We now write:

The $RO_2$ formed in (R1) and (R4a) do not form stable peroxy-nitrates in their reaction with $NO_2$, so this $RO_2$ loss process can be safely neglected.
* * *
5. Page 8, line 8: Clarify to say that the ANs production from ozonolysis has a daytime minimum at noon (since the absolute minimum is really at night).

Corrected. We now write:

The rate of production of ANs from ozonolysis of BVOC has a daytime minimum at noon, with maximum values observed in the late afternoon.
* * *
6. Figures 2 and 3: x-axis labels are confusing.

X-axes given as dd/mm
* * *
7. Page 9, line 15: I think your estimate uses the "steady-state" approximation rather than the "stationary-state" approximation.

Correction made.
* * *
8. Figure 6: Please define what your error bars are (standard deviation?). Additionally, the ends of some of the error bars are not visible in the plot. Is the fit you are doing to all points or only the average points that are plotted? What kind of fit are you using (OLS, RMA, York?)?

We have redrawn this Figure and added x-axis errors. We have used a York-type fit and now write:

Within the overall uncertainty represented by the error-bars, there is no significant difference between the day- and night-time data, with a linear fit through all the data indicating a lifetime of $\approx 2 \pm 3$ hours.

The caption mentions the fit-type:

The slope of the linear fit to the data (York-fit, errors in both axes considered, black line) indicates a lifetime of $2 \pm 3$ h.
* * *
9. Page 9, line 25: Some alkyl nitrates (e.g., isoprene hydroxy nitrate) can be oxidized by OH with reasonable efficiency, and highly oxidized or carbonyl nitrates can be rapidly photolysed (see Muller et al., 2014 and Xiong et al., 2016). Are these not relevant during IBAIRN?

The studies mentioned deal with isoprene derived ANs, which were not important for IBAIRN. We clarify this by writing:

ANs are generally thought to react inefficiently with $O_3$, OH and $NO_3$ and low rates of photolysis mean that their lifetimes are likely to be controlled largely by dry deposition and / or heterogeneous hydrolysis on aerosol or hydrometeors (Browne et al., 2013). Known exceptions are some ANs formed from isoprene, which can react with OH and/or be photolysed with lifetimes on the order of an hour (Muller et al., 2014;

Xiong et al., 2016). During IBAIRN, isoprene derived ANs were however only a small fraction of the total and, in the absence of kinetic / photochemical data for terpenes, we disregard gas-phase, chemical loss processes.
* * *
10. Page 9, line 32: Should be "assessed" instead of "accessed."

Correction made.
* * *
11. Page 10, line 3: Was the calculation of average thermal velocity done at STP or at the average temperature and pressure during the campaign? Please specify.

We have amended the text and now write:

For a C10 alkyl nitrate derived from monoterpene oxidation such as $C_{10}H_{14}NO_7$, (Yan et al., 2016; Lee et al., 2018) $\bar{c}$ ~15000 cm s$^{-1}$ at 290 K.
* * *
12. Page 10, line 11: Consider citing Romer et al., 2016 and Zare et al., 2018 which also discuss ANs lifetimes and heterogeneous hydrolysis as a loss pathway for ANs.

Citations included.
* * *
13. A separate conclusion section would be helpful to the reader (i.e. add a section header before the last two paragraphs).

We have added a section header "conclusions" and reorganized the text somewhat.

**4 Conclusions**

During the IBAIRN campaign in the boreal forest in southern Finland (5$^{th}$-22$^{nd}$ Sept, 2016), alkyl nitrate formation was dominated by the reaction of $NO_3$ radicals with monoterpenes, both during the day- and night-time, with smaller contributions from both OH and $O_3$ initiated oxidation of BVOCs. This result highlights the important role of daytime $NO_3$ chemistry (with respect to organic nitrate formation) in this environment. The short, average lifetime of ≈ 2 h for the total alkyl-nitrates (ΣANs) indicates efficient uptake to existing particles and/or deposition.

These observations, of efficient daytime production of gas-phase ANs from $NO_3$ chemistry and short night-time lifetimes are entirely consistent with the results from recent studies at the IBAIRN site by Lee et al. (2018) who found that organic nitrates previously designated as resulting from night-time processing of BVOCS (Yan et al., 2016) were also present during daytime. In addition, they found relatively few organics with "night-time" character in the gas-phase compared to the aerosol-phase, indicating efficient transfer of gas-phase organic nitrates to the particle-phase at night-time, likely aided by low temperatures and high relative humidity. We found no significant change in the production rate of ANs in transition from summer

to autumn, though the short duration of the campaign and variability in temperature and insolation would mask such effects.

---

## Author Comment (AC2)

**Referee 2**

In the following, the referee's comments are reproduced (black) along with our replies (blue) and changes made to the text (red) in the revised manuscript.

**General Comments:**

This concise and clear paper reports on measurements of alkyl nitrates in the boreal forest, with coincident measurements of organic trace gases enabling an assessment of the relative source strength of OH, NO3, and O3 oxidation in producing these alkyl nitrates. In this NOx-limited environment, NO3 oxidation is found to be the dominant source of alkyl nitrates both night and day. The paper is clearly written and the figures are helpful. I suggest addition of a bit more auxiliary data to enable readers to better interpret the conclusions.

We thank the referee for this positive assessment of our manuscript.

\_\_\_\_\_

1) As I read this paper and sought to understand the key observations, I found myself wondering about the [NO] and relative concentrations of different organic trace gases. These data are perhaps in other papers cited, but for convenience of the reader I urge the authors to include this data here. I suggest to include an NO trace in the top panel of Figure 2, and add a panel to that figure showing BVOC timeseries, perhaps split out by isoprene and summed terpenes since they likely have different diel patterns, and since their relative reactivities is different and can help the reader interpret the day/night observations.

We have followed these suggestions and added mixing ratios of NO and the terpenoids to the plot.

\_\_\_\_\_

2) For similar reasons, it would be helpful to add another panel to the diel average figure 4, showing NO2 and BVOC traces. In particular, I was curious why the NO3-initiated production of ANs would peak at 19:00 local time and then decrease? Given your statement that monoterpene concentrations build up overnight, I might have expected this to continue increasing. Is it that the NO2 is fully consumed by then? We have added the requested plots (Fig 4b). The reason for the reduction in the ANs production rate after 20:00 is lower  $NO_2$  and  $O_3$  mixing ratios. We have added text to explain this:

The peak in the night-time production rate of ANs at 19:00 coincides with large O3 and NO2 mixing ratios (Fig 4b), the reduction of both between ~ 20:00 and mid-night (UTC) resulting in the decrease in  $\sum P_{ANs}^{NO_3}$  during the night, though changes in relative concentrations of the terpenes may also play a role.

\_\_\_\_\_

3) I agree with Jacqui, Figure 5 is not necessary.

Figure 5 summarises (perhaps better than words can) one of the major findings of this study, that  $NO_3$  reactivity in this region/season is so high that it contributes to ANs production not only at night-time, but

also during the day. The plot does not take up much space we would prefer to keep it as Figure and graphical abstract.
* * *
4) Abstract line 22: "strongly controlled by biogenic emissions" – ? seems inconsistent with your discussion in the manuscript body. There, you describe this as due to rapid deposition to particles?
We have re-worded this part of the abstract.

The lifetimes of the gas-phase ANs formed in this environment were of the order of 2 hours due to efficient uptake to aerosol (and dry-deposition), resulting in the transfer of reactive nitrogen from anthropogenic sources to the forest ecosystem.

\_\_\_\_\_

5) Around p. 3 line 23-24: Could you list the dominant terpenes here? (and / or, on p. 5 around line 13 where you state that only a handful contributed significantly to reactivity – include a brief ranked list?) Also, what anthropogenic emissions are observed from the cities & sawmill – just NOx, or NOx and SO2? In response to a suggestion of Referee #1, we have added the following text:

A large selection of VOCs was measured (a listing is given in the caption to Figure S2 of the supplementary information) but the 5 biogenic VOCs listed ( $\alpha$ -pinene,  $\beta$ -pinene, carene, limonene, isoprene) accounted for > 98 % of the attributed NO3 reactivity.

From the sawmill we saw an increase in BVOCs. From the cities an increase in  $NO_X$  (not  $SO_2$ ). This is now mentioned:

Anthropogenic emissions from two larger cities (Tampere and Jyväskylä) and a local sawmill occasionally impacted the site, the former resulting in an increase in NOX levels, the latter in BVOCs.

\_\_\_\_\_

6) On p. 8 around line 18: say something about why the NO3 initiated formation of ANs peaks at 19:00 See comment 2. We have added a Fig. 4b and the following text:

The peak in the night-time production rate of ANs at 19:00 coincides with large O3 and NO2 mixing ratios (Fig 4b), the reduction of both O3 and NO2 between ~ 20:00 and mid-night (UTC) resulting in the decrease in  $\sum P_{ANs}^{NO_3}$  during the night, though changes in relative concentrations of the terpenes may also play a role.
* * *
7) On Figure 6, the error bars on the bottom panel look very large compared to the reported slope uncertainty of  $\pm 0.5$  hr. Please explain how this error bar is determined – it looks to me like the slope could even be negative within the uncertainties.

The uncertainty was calculated in a weighted linear fit considering errors in the y-axis data only. The quoted uncertainty was correct.

However, we have now reconstructed Fig. 6 with x-axis errors and carried out a York fit to the data (errors in both x- and y-axes considered). The slope is now quoted as  $2 \pm 3$  h. The new lifetime is listed in the Figure caption and in the text.

\_\_\_\_\_

8) Figure 7 makes me wonder whether it's possible that different sensitivities of I- CIMs to daytime vs. night-time BVOC mixes could explain the different amplitude of the diel cycle. Can you add anything additional information on this?

As we already write, neither the absolute nor the relative sensitivity (between different alkyl-nitrates) is known and we cannot examine this aspect in detail. However, we have added a sentence to indicate that this may contribute to the disparate profiles in Fig. 7:

In addition, we cannot rule out that this difference is due to different HR-L-ToF-CIMS sensitivity to dayand night-time ANs.
* * *
**Technical corrections/suggestions:**

p. 2 line 21: "OH radicals are largely absent"

We now write: At night-time, OH radical concentrations are very low

line 27: "Reaction 6a is a composite". Correction made.

**p. 3** line 10: suggest to add reference to Ayres 2015 (https://www.atmos-chemphys.net/15/13377/2015/):

this NO3 + BVOC dominance during the day was also observed at SOAS 2013. Reference added.

p. 3 line 30 "reached 100% during many nights" end. Correction made.

p. 3 / top of p. 4: Is 300 pptv the average NOx level for the whole campaign? Maybe also mention the [NOx] during the events where air masses arrive from the industrial sources.

We now write: The NOx levels during the entire campaign were low (mean value of 320 pptv) with occasional increases (up to 1.4 ppbv) when the site experienced air masses with trajectories that passed over urban centres.

**p. 4** line 4 "photolysis frequency, and the" Correction made.

line 21: remove extra ")" Correction made.

lines 25-26: "OH concentrations have an associated uncertainty of ~ 50%" Correction made.

line 33: add citation for I- CIMS high sensitivity to nitrates. Citation (Lee et al, 2016) added.

**p. 5** Eq. 2: It's a little confusing that you use the average alpha in the equation but then talk about the individual ones first below the equation, and then define the average in Eq 3 below. Maybe combine Eq. 3 into 2 so you see the average and the summation simultaneously? Also, after the current Eq. 3, define the Ci term.

We have reorganized this and now write:

where  $\overline{\alpha}^{NO_3}$  is an average AN-yield. Assuming that all the VOCs responsible for loss of NO3 were identified and quantified the average yield can be derived (Eq. 3) from VOC- specific values of  $\alpha_i^{NO_3}$  weighted by their relative contribution to  $k_{OTG}^{NO_3}$ .

$$\overline{\alpha}^{NO_3} = \frac{\sum \alpha_i^{NO_3} k_i^{NO_3} [C_i]}{k_{\text{OTG}}^{NO_3}}$$
(3)

where  $[C_i]$  is the concentration of the specific VOC.

Eqn 4: k3 should be k5? Correction made.

Line 24 "total measured reactivity". Correction made.

**p. 6** Eqn R12: meaning of the "delta" term is unclear.  $\Delta$  has been replaced with the word isomerization. line 27: "UTC). In order to account for this competition with HO2 reactions, equation (7) can be modified to:" Correction made.

**p. 7** line 3: "the local sawmill, likely due to elevated reactivity with ....?" We now write: was associated with large BVOC mixing ratios in air masses originating from the local sawmill.

Line 4-5: This sentence sounds like you're drawing a contrast to NO3, but I think this is true in that case as well. Perhaps make this sentence the first sentence of the next paragraph instead? Text re-organised Line 11: "We show below that even if unattributed OH-reactivity reaches 50%, this would not

significantly" Text changed as suggested.

Line 13: insert space "from [Ci]" Correction made.

Line 20: "Similarly as for OH-reactions," We write: Similar to OH reactions.

**p. 8** line 2: "was estimated to be ~ 60%" Correction made.

line 24: "ANs production rate occurs exclusively via NO3-initiated reactions." Correction made.

p. 9 line 1 "which was 570 m" Correction made.

line 2: omit extra ")" Correction made.

line 7: "which is not the case (Eerdekens" Correction made.

line 20: "overall uncertainty represented by error bars, there" Correction made.

line 27: "well-mixed daytime boundary" Correction made.

line 28: include units on Vdep (here and in the line below "Vdep ~ 2") Units included.

line 32: "can be assessed" Correction made.

p. 10 line 3: "For typical alkyl nitrate" Correction made

line 18 & below: Shouldn't the CIMS be designated the "I - CIMS" and not "I-CIMS"? We now name it the HR-L-ToF-CIMS as in section 2

p. 11 top 2 lines: C9 shows up in two categories ? Corrected, C9 now appears only once